# A Design of Experiment Approach for Development of Electron Beam Powder Bed Fusion Process Parameters and Improvement of Ti-6Al-4V As-Built Properties

**Dor Braun [1,2], Yaron Itay Ganor [2,3], Shmuel Samuha [3], Gilad Mordechai Guttmann [3], Michael Chonin [2], Nachum Frage [1], Shmuel Hayun [1]** 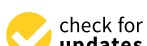 **and Eitan Tiferet [2,3,*]**

[1] Department of Materials Engineering, Ben-Gurion University of the Negev, Beer Sheva 8410501, Israel
[2] Additive Manufacturing Center, Rotem Industries Ltd., Rotem Industrial Park, Mishor Yamin, DN Arava 8680600, Israel
[3] Department of Materials, Nuclear Research Center Negev, Beer Sheva P.O. Box 9001, Israel
[*] Correspondence: eitant@nrcn.gov.il

**Abstract:** Additive manufacturing is a novel and breakthrough technology by which parts can be manufactured for various purposes and services. As in any production process, the desired properties of additively manufactured components, particularly in electron beam melting processes, ultimately depend on the manufacturing process parameters. Process parameters should be designed accordingly to manufacture parts with specific and desired characteristics. This study focuses on examining the effect of process parameters, such as beam current and velocity, focus offset, and line offset, at three different values each, on the properties of Ti-6Al-4V alloy. The study on the effect of the process parameters on the as-built material's performance was performed using the Taguchi approach using an $L_9$ ($3^4$) orthogonal array. The properties of printed parts (density, surface roughness, elastic moduli, hardness, tensile characteristics, fractography, and microstructure) were tested. A wide range of properties was obtained and analyzed; namely, porosity varied from 8% to almost fully dense materials with density higher than 99.9% and a range of yield and ultimate tensile strength values and brittle samples with less than 1% elongation to ductile samples with an elongation greater than 16%. The overall performance of printed parts was determined based on an evaluation criterion. Several parameter combinations were found and yielded the fabrication of parts with high density and relatively fine microstructure. The comparison of the best parameter combinations determined in this study and the parameters recommended by the machine manufacturer showed that improved results were obtained, and even when using the optimal parameters, they can be improved even more. This result highlights the ability of the proposed DOE method to further develop existing results and even for development of manufacturing parameters for new materials.

**Keywords:** electron beam melting; mechanical properties; microstructure

## 1. Introduction

Ti-6Al-4V (known as Ti64) is a dual-phase alloy, namely an α-Ti (HCP) and β-Ti (BCC) alloy, due to the presence of aluminum and vanadium, α and β stabilizers, respectively, with small impurities of other elements (ppm). Since Ti64 is an α + β alloy, the microstructure and thus the mechanical properties depend upon the processing history and heat treatment. The alloy microstructure is determined by the β→α transformation that occurs during cooling. Depending on the cooling rate, the microstructure can occur by nucleation and growth to form the well-known Widmanstätten (also known as "basket-weave") microstructure or martensitic (α' or α'') phase. This alloy is the most widely used among titanium alloys [1]. It exhibits high strength, stiffness, and toughness and low density, making it extremely attractive for aerospace applications, such as airframe and jet engine components [1,2]. The excellent corrosion resistance and high strength make it useful in many other fields,

such as marine, automobile, energy, chemical, and petrochemical. The fact that it exhibits superior biocompatibility extends its usage even to biomedical industries for implants and prosthetics devices [3,4].

AM, widely known as 3D printing, is defined as the manufacturing process of building three-dimensional (3D) objects by slicing a 3D model into slices, then producing each slice while adding material to previous layers, joining each slice to the previous one, as opposed to conventional subtractive manufacturing, such as traditional machining [5,6]. Out of all the many AM technologies, one of the main and most common manufacturing methods is powder-bed fusion (PBF). The manufacturing process using this method has been described extensively and in detail in previous studies [4,7–9]. Various materials can be used in PBF, such as polymers, ceramics, and even composites, but is used primarily for metals [5]. For metals manufacturing, the two methods differ from each other mainly by the type of energy source, i.e., laser (L-PBF) and electron beam (EB-PBF), and as a result sets the atmosphere of the process, i.e., inert gas (commonly argon) and controlled medium vacuum, respectively [10]. Another difference between the methods is the process temperature; while in L-PBF, the process takes place at a temperature below 250 °C, in EB-PBF, the process is carried out at elevated temperatures [11–13]. These differences affect the properties of the parts obtained; L-PBF parts are characterized by a finer microstructure resulting in higher tensile strength and better fatigue resistance but lower ductility [14,15]. On the other hand, these systems' differences grant several advantages to EB-PBF over L-PBF; the final product is characterized by very low residual stresses, highly reactive material can be used without significant oxidation, and it also produce multiple components within a single process by stacking parts on top of each other.

Previous studies in our group have investigated the dependence of mechanical properties on location in the build chamber and the resulting variance and the resulting texture dependence on the tilt angle from the build direction and the direction of the heat dissipation [16–20]. All of these were performed under the production parameters recommended by the manufacturer. In addition, the effect of post-process heat treatments on the properties was investigated in order to improve the properties obtained using the standard manufacturing parameters [21]. Usually, the properties improvement through post-processing comes with a price that reduces other properties. Therefore, there is motivation to learn to improve the final properties without the need for post-processing.

EBM process depends on many parameters, some of which characterize the beam, such as the beam velocity, current, and diameter. Other parameters, such as scan strategy, ambient temperature, hatch spacing, or average energy input, can also affect produced parts. Among these many parameters, the most important and influential for the melting step are the parameters that control the energy density deposited into the powder layer: beam current, velocity, focus offset, and line offset. Adjustment of these parameters affects the energy density deposited to the powder bed. Consequently, it impacts the melt pool size and depth, thermal distribution, and cooling rate, which correspondingly affects the solidification process, crystallographic textures, and microstructure morphology. Thus, the final microstructure and consequently the mechanical properties depend significantly on the process parameters. Therefore, by changing the process parameters, it is possible to obtain a variety of properties for the same material within the same technology, machine, and even the same build [4,22]. Since the goal is to form fully dense parts, sufficient energy must be provided to melt the entire desired area. While lack of energy can cause defects such as lack-of-fusion porosity and delamination, excessive energy can cause defects such as keyhole, balling, swelling, and cracks [8,23,24].

Many studies have been conducted on the effect of manufacturing parameters on the properties of the material obtained, but most have examined the effect of individual parameters or based on the creation of process maps (or process windows) [25–35]. Although these approaches make it possible to obtain useful information regarding trends of individual parameters and efficient and appropriate specific parameter combinations, a change in one of the parameters set out in the studies can result in dramatic differences in the resulted

trends and outcomes. Moreover, since the EB-PBF process includes thermal management, the differences between the various EB machines greatly impede the application of the results obtained from one machine to another. The Arcam EB-PBF Q20Plus machine is characterized by a large chamber and manufacturing volume that affects the thermal experience of each part and the overall thermal history that occurs during the production process; therefore, it falls to the manufacturing parameters to adjust and compensate throughout the build process.

The formation of process windows is based on the full factorial design of experiments (DOE). Since it is obtained by trial and error, full factorial is a costly and time-consuming method for examining many parameters. One way to investigate the results of several parameters is using a statistical DOE, in which only a fraction of the full factorial number of combinations is performed. Taguchi method is a fractional factorial DOE that minimizes the number of experiments required, allowing significant savings in time and costs. It is a powerful statistical tool for improving the performance of products at the development stages. Due to its efficiency, the Taguchi method gained vast popularity, and therefore, it is used in many engineering fields [36–42]. Using the Taguchi method, it is possible to study the effect of several parameters over a wide range of values simultaneously and without the need to produce a large number of samples. Accordingly, this study utilized the Taguchi method using an orthogonal array (L9) to examine multiple parameters. This study presents a DOE method and study results of process parameters development in EB-PBF on an Arcam Q20+ machine for improvement of bulk properties of Ti64 alloy. The study aims to offer a DOE method to find a parameter set that can yield the best results even when developing new parameter sets for new materials.

## 2. Experimental Methodology

### 2.1. Design of Experiment

EB-PBF requires consideration of multiple process parameters before manufacturing (e.g., layer thickness, beam current, focus offset, line offset, speed function index, line order, preheat current, contours, etc.) [11]. Each of these parameters affects the quality of the manufactured product; however, the examination of the effect of all these parameters on the process is not realistic due to time and cost constraints. Of all the parameters that affect the EB-PBF process, this study selected four main parameters that control the melting characteristics and examined their effect on the final manufactured components. Each parameter was assigned three levels (values) based on Arcam's standard melt theme and taking into consideration the overall energy deposition. The chosen parameters and their levels are presented in Table 1. Since the deposited energy depends on each tested parameter, a working window was chosen that allows obtaining samples from each parameter combination but, on the other hand, to enable distinctive variation. It is worth mentioning that these parameters affect only the hatching stage in the manufacturing process and not the contour of the manufactured parts. In a full factorial, this system consists of four parameters and three levels, and the number of possible combinations is 81, which would be very costly to manufacture and measure. In addition, in the case that a number of replications is needed for increased accuracy, the total amount of samples is vast and would require copious amounts of resources and time. Therefore, the Taguchi method was applied to minimize the number of experiments. For this kind of experimental system, an orthogonal array (OA) $L_9$ ($3^4$) was used through which only nine experiments are required. The design of the experiment is shown in Table 2.

**Table 1.** Process parameters and their levels.

| Parameters \ Levels | 1 | 2 | 3 |
|---|---|---|---|
| Beam current (I) (mA) | 25 | 28 | 30 |
| Focus offset (FO) (mA) | 15 | 45 | 75 |
| Line offset (LO) (mm) | 0.15 | 0.22 | 0.3 |
| Speed function index (SF) | 20 | 32 | 45 |

**Table 2.** Experimental design as per L9 orthogonal array. Each row represents a parameter set.

| Experiment # | I (mA) | FO (mA) | LO (mm) | SF (AU) |
|---|---|---|---|---|
| 1 | 25 | 15 | 0.15 | 20 |
| 2 | 25 | 45 | 0.22 | 32 |
| 3 | 25 | 75 | 0.3 | 45 |
| 4 | 28 | 15 | 0.22 | 45 |
| 5 | 28 | 45 | 0.3 | 20 |
| 6 | 28 | 75 | 0.15 | 32 |
| 7 | 30 | 15 | 0.3 | 32 |
| 8 | 30 | 45 | 0.15 | 45 |
| 9 | 30 | 75 | 0.22 | 20 |

## 2.2. Sample Preparation

Samples were manufactured using the EB-PBF process in an Arcam Q20plus machine (Arcam EBM, Mölndal, Sweden). Except for the variation in beam current, focus offset, line offset, and speed function index, all other process parameters were under the standard parameters recommended by the manufacturer. The manufacturing process was carried out in a vacuum atmosphere of $4 \times 10^{-3}$ mbar and at a temperature of 700 °C. First, 45–106 μm Ti6Al4V ELI plasma atomized powder, in compliance with the ASTM F3001 [43], was supplied by Arcam. The particle size distributions for D10, D50, and D90 values were 51, 69, and 103 μm, respectively. Each set of parameters was used to manufacture five samples (sample set) to increase accuracy. A total of 45 cylindrical specimens with cubes on top were produced (Figure 1). The samples were cut to separate the cubes and cylinders. The top part of the cube was cut for surface roughness measurements, while the rest was machined for density, ultrasonic, XRD, and hardness measurements. Tensile samples were machined from the center of the cylinders.

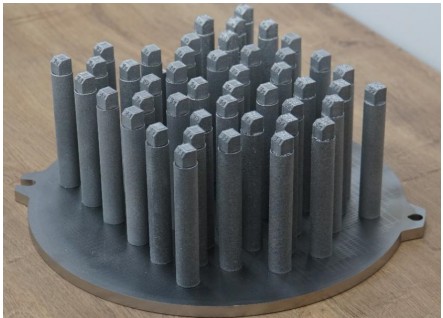

**Figure 1.** 45 Ti-6Al-4V samples tray.

## 2.3. Sample Characterization

Characterization was performed by measurements of density, longitudinal and transverse sound wave velocity, tensile strength, hardness, and surface roughness and also included fractography and microstructure analysis. The density measurements were carried out using the previously described Archimedes method [44]. Six pulse-echo ultrasonic

measurements were conducted along each sample's build direction (*z*-axis) for both longitudinal and transverse time-of-flight (TOF) measurements. Then, 5 MHz probes were used coupled with commercial couplant material SWC (Shear wave commercial couplant, by Olympus). The longitudinal and transverse sound wave velocities were calculated using the sample thickness and the average TOF in each longitudinal and transverse direction. Tensile tests were performed parallel to the build direction using a Zwick 100 Allround electro-mechanical screw tensile machine per the ASTM E8/E8M standard [45]. All tests were position-controlled with a crosshead travel velocity of 1 mm per minute. The strain was measured using a 25 mm clip-on extensometer up to 2% elongation. Vickers hardness measurements were performed under a 0.5 kgf load and 10 s dwell time using a Buehler Micromet (Lake Bluff, IL, USA) in accordance with the ASTM E92 standard [46]. The average hardness values of each parameter group were measured by selecting three samples from each parameter group and performing eight measurements per sample. The surface roughness was measured using a non-contact chromatic confocal optical profilometer Nanovea PS50 (Nanovea Inc., Irvine, CA, USA). Five line profile measurements were conducted per sample using a 300 nm optical pen. The lines' length and step sizes were 3 mm and 1 µm, respectively. The beam frequency was 100 Hz. Three selected samples were sampled from each parameter set, and the average roughness ($R_a$) and the average maximum profile height ($R_z$) were calculated. Fractography and microstructure observations were made using FEI Verios XHR 460 L scanning electron microscope (SEM) (Hillsboro, OR, USA).

## 3. Results and Discussion

The elastic moduli, shear modulus, Young's modulus, bulk modulus, and Poisson's ratio, *G*, *E*, *B*, and *ν*, respectively, were calculated based on the measured longitudinal and transverse sound wave velocities using equations 1–4 [47,48]. Table 3 summarizes the calculated elastic moduli values. It can be seen that samples in set 3 exhibit the lowest *B*, *E*, and *G* values. With regards to the rest of the samples sets, except for sample set 6, in which wide value ranges were obtained, the values of the different elastic moduli, *B*, *G*, *E*, and *ν* values, are in the range of 108–114 (GPa), 44–46 (GPa), 118–121 (GPa), and 0.31–0.33, respectively.

$$G = \rho V_t{}^2 \tag{1}$$

$$E = \rho V_t{}^2 \left( \frac{3V_l{}^2 - 4V_t{}^2}{V_l{}^2 - V_t{}^2} \right) \tag{2}$$

$$B = \rho \left( V_l{}^2 - \frac{4}{3} V_t{}^2 \right) \tag{3}$$

$$\nu = \frac{1}{2} \left( \frac{V_l{}^2 - 2V_t{}^2}{V_l{}^2 - V_t{}^2} \right) \tag{4}$$

where $\rho$ is the density, and $V_l$ and $V_t$ are the longitudinal and traverse velocities, respectively.

Table 4 presents the density measurements as well as the relative density, while theoretical density was taken as 4.43 (g/cm³). These are the average results from five samples and five measurements per sample. The density ranged from 92.01 to 99.86%, where samples fabricated under parameters sets 3 and 6 exhibited the lowest density, while samples sets 5 and 7 showed the highest ones. Except for samples sets 3 and 6, the relative density of all other samples is over 99.63%. The relative density of the samples sets is shown in Figure 2. The small error bars (except for samples 3 and 6) indicate uniformity between the samples from the same set of parameters, which indicates the repeatability of the process. A previous study found that differences in hundredths of percent in relative density showed variation in mechanical properties [17]. Therefore, the samples were classified into four groups according to the relative density, as seen from the color coding in Figure 3, which presents the sample's top surface. On the top surface of samples 3 and 6, the porosity is apparent and clear. This type of surface indicates insufficient melting due to a combination of incompatible manufacturing parameters [8,33] causing a lack of fusion

defects, such as unmelted powder residues and pores, that decrease the sample density and are in accordance with the low-density results obtained (Table 4). However, a waviness can be noticed from the surface of sample 1, which indicates over-melting. Apart from the above samples sets, the other sets exhibit a flat surface and neat melting tracks that indicate suitable production parameters consistent with the density results.

**Table 3.** Elastic moduli values of the various samples.

| Sample Set # | Bulk Modulus, K (GPa) | Young's Modulus, E (GPa) | Shear Modulus, G (GPa) | Poisson's Ratio, $\nu$ |
|---|---|---|---|---|
| 1 | 109.38 ± 0.61 | 120.07 ± 0.46 | 45.58 ± 0.23 | 0.3172 ± 0.0016 |
| 2 | 110.99 ± 0.65 | 120.37 ± 0.4 | 45.62 ± 0.2 | 0.3192 ± 0.0013 |
| 3 | 60.47 ± 5.27 | 62.69 ± 2.49 | 23.64 ± 1.06 | 0.3264 ± 0.015 |
| 4 | 108.68 ± 0.41 | 120.61 ± 0.16 | 45.86 ± 0.07 | 0.315 ± 0.0007 |
| 5 | 111.33 ± 0.61 | 118.36 ± 0.23 | 44.74 ± 0.1 | 0.3226 ± 0.0013 |
| 6 | 113.58 ± 7.52 | 106.76 ± 8.16 | 39.81 ± 3.63 | 0.3422 ± 0.0214 |
| 7 | 108.32 ± 0.4 | 121.02 ± 0.21 | 46.06 ± 0.07 | 0.3138 ± 0.0004 |
| 8 | 110.53 ± 0.3 | 121.07 ± 0.31 | 45.95 ± 0.14 | 0.3172 ± 0.0008 |
| 9 | 113.08 ± 1.16 | 118.83 ± 1.61 | 44.85 ± 0.72 | 0.325 ± 0.0037 |

**Table 4.** Density, relative density, yield strength, ultimate tensile strength, elongation at break, and hardness values.

| Sample Set # | Density (g/cm$^3$) | Relative Density (%) | $\sigma_y$ (MPa) | $\sigma_{max}$ (MPa) | $e_f$ (%) | Hardness (HV$_{0.5}$) |
|---|---|---|---|---|---|---|
| 1 | 4.4202 ± 0.0026 | 99.78 ± 0.06 | 887.3 ± 4.1 | 955.3 ± 13.7 | 7.7 ± 4.0 | 385.1 ± 14.9 |
| 2 | 4.4136 ± 0.0011 | 99.63 ± 0.02 | 937.7 ± 12.2 | 1013.3 ± 11.5 | 13.7 ± 3.0 | 383.9 ± 17.5 |
| 3 | 4.0761 ± 0.0179 | 92.01 ± 0.40 | 360.4 ± 50.4 | 383.6 ± 49.1 | 0.9 ± 0.1 | 381.7 ± 18.8 |
| 4 | 4.4182 ± 0.0027 | 99.73 ± 0.06 | 908.7 ± 5.5 | 983.4 ± 6.6 | 14.7 ± 0.6 | 380.7 ± 14.1 |
| 5 | 4.4240 ± 0.0115 | 99.86 ± 0.26 | 892.7 ± 8.7 | 976.0 ± 8.5 | 15.1 ± 1.5 | 371.67 ± 13.2 |
| 6 | 4.2192 ± 0.0368 | 95.24 ± 0.83 | 705.5 ± 57.5 | 738.4 ± 51.9 | 1.6 ± 0.2 | 394.6 ± 15.9 |
| 7 | 4.4234 ± 0.0011 | 99.85 ± 0.02 | 899.6 ± 4.5 | 970.9 ± 5.7 | 14.6 ± 1.7 | 383.9 ± 14.0 |
| 8 | 4.4144 ± 0.0009 | 99.65 ± 0.02 | 978.6 ± 11.1 | 1050.1 ± 8.4 | 14.2 ± 0.9 | 379.1 ± 10.9 |
| 9 | 4.4184 ± 0.0014 | 99.74 ± 0.03 | 930.4 ± 9.0 | 1021.3 ± 6.4 | 16.9 ± 0.7 | 371.5 ± 10.6 |

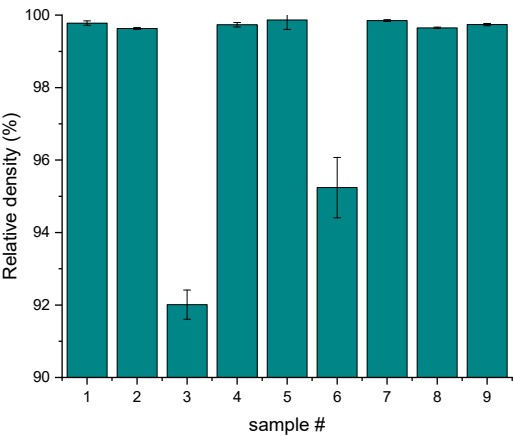

**Figure 2.** Relative density of samples.

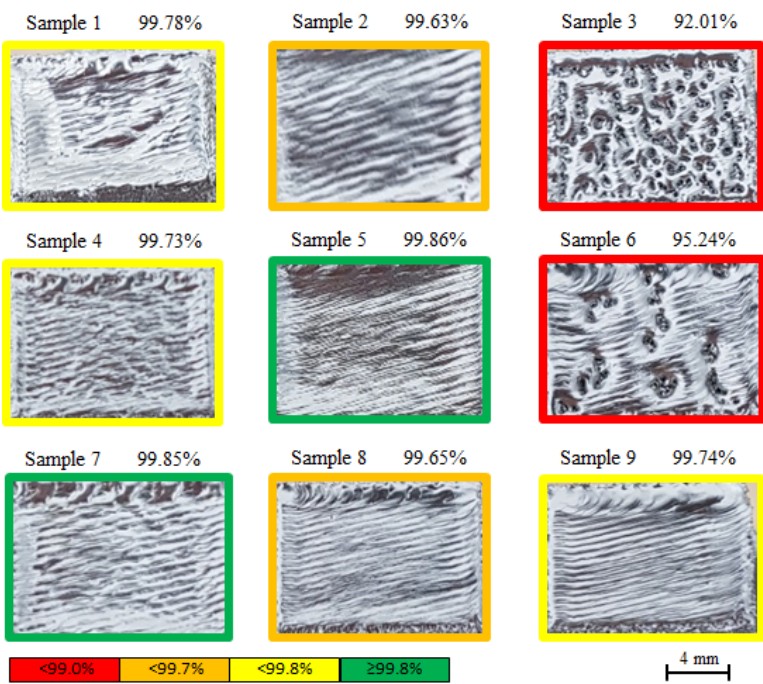

**Figure 3.** The top surface of representative samples from each set.

The top surface of representative samples from each set of samples is shown in Figure 3. As seen in the Figure 3, the top surface of samples 3 and 6 are rough and coarse and therefore were not measurable. Additionally, the surface of samples manufactured using parameter set 1 was not measurable due to the significant (more than 0.5 mm) curvature. For the rest of the sample sets, the average results from three samples and five measurements per sample are shown in Table 5. As can be seen from the Table 5, sample set 9 generated the smoothest surface both in terms of arithmetic mean deviation of the profile ($R_a$) and in terms of the average maximum peak to valley height of the profile ($R_z$). Sample sets 2, 5, and 8 have a slightly rougher surface than those in sample set 9. Sample sets 7 and 4 have the roughest surface of all the samples measured, while sample set 4 has the highest $R_a$ values, and sample set 7 has the highest $R_z$ values. Figure 4 graphically depicts the surface roughness results with reference to the FO value through which each sample set was manufactured. Figure 4 shows that the surface roughness distribution corresponds to the FO parameter. However, it is important to bear in mind that there are samples that were not measurable: samples sets 1, 3, and 6, which were produced using a FO of 15, 75 and 75 mA, respectively. Therefore, although it is clear from Figure 4 that the use of 75 mA reduces the surface roughness, in practice, two samples manufactured using this value were so coarse that they could not be measured. Nevertheless, despite the difference in results between the samples sets, compared to the roughness results found in the literature, ranging from 6 μm to even 15.8 μm [49–51], the roughness obtained is relatively low.

**Table 5.** Surface roughness results.

| Sample Set | 2 | 4 | 5 | 7 | 8 | 9 |
|---|---|---|---|---|---|---|
| Ra (μm) | 2.14 ± 1.04 | 3.49 ± 0.80 | 2.09 ± 0.52 | 3.31 ± 1.04 | 2.02 ± 0.61 | 1.62 ± 0.26 |
| Rz (μm) | 12.78 ± 4.83 | 19.65 ± 3.10 | 13.50 ± 4.55 | 22.39 ± 6.00 | 14.03 ± 4.68 | 11.13 ± 1.38 |

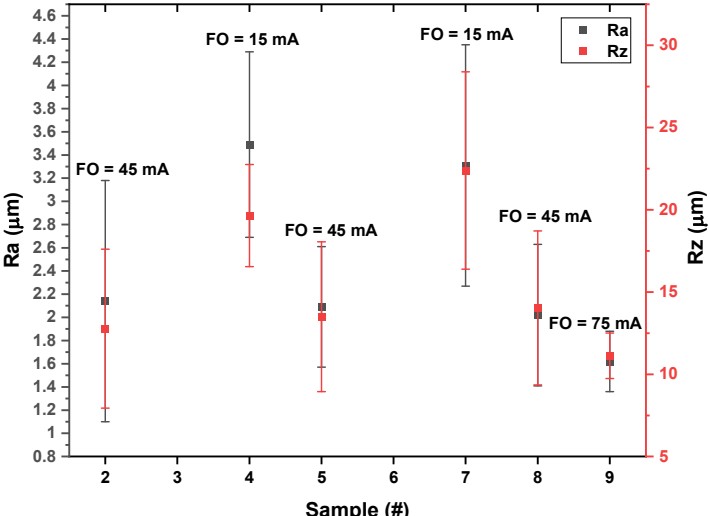

**Figure 4.** Average surface roughness for each sample set and the focus offset (FO) used.

The mechanical properties, yield strength (YS), ultimate tensile strength (UTS), and elongation at break ($e_f$) are shown in Table 4 and are plotted graphically, as can be seen in Figure 5. Figure 5 conveniently highlights the narrow scattering obtained for the strength of each samples sets, indicative of the process repeatability. In terms of elongation, the variance is slightly larger but still within reasonable limits. It can be seen that the lowest YS, UTS, and $e_f$ values measured are from the sample sets 3 and 6, similar to the density results obtained. On the other hand, the highest YS and UTS obtained are from sample sets 2, 8, and 9, and the sample set that presented the highest $e_f$ is 9. From comparing the strength results to the density results, it can be seen that the highest strength is not strongly correlated with the highest density. While density is a parameter that has a critical effect on the material's properties (i.e., samples 3 and 6 where the density and the resulting strength are low), the effect does not extend to sets where the density was high (above 99.6%), and the differences were relatively small (~0.2%).

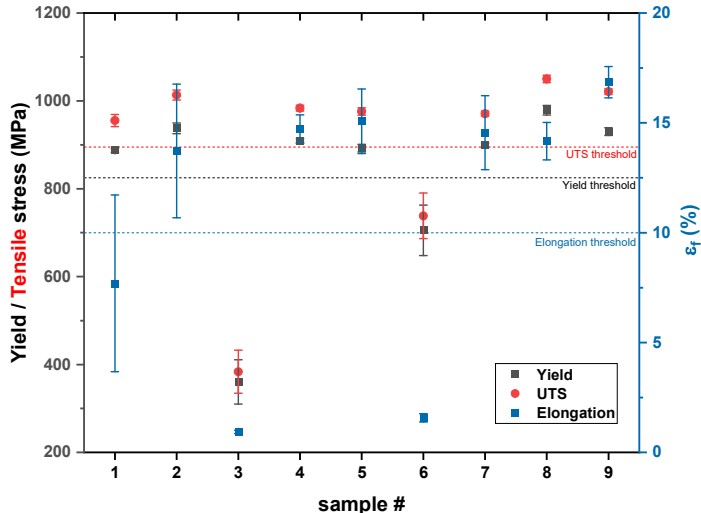

**Figure 5.** Mean values and standard deviations of yield strength, ultimate tensile strength, and elongation. The horizontal dashed lines are the yield (black), UTS (red), and elongation (blue) thresholds according to ASTM F2924 [54].

Comparing the YS and UTS results to the values required by ASTM standards, it can be seen that except for sample sets 3 and 6, all sample sets meet the required values in the

standards for cast material, wrought material, and AM-PBF material (ASTM F1108 [52], ASTM F1472 [53], and ASTM F2924 [54], accordingly). Regarding the elongation results, all samples except sample sets 1, 3, and 6 meet the requirements of ASTM standards for the cast, wrought, and AM-PBF material.

The hardness results are also presented in Table 4, and it can be seen that there is no correlation between them and the YS, UTS, and density. Samples with relatively low YS, UTS, and density values showed relatively high hardness (samples sets 6 and 3). In contrast, the samples with the highest strength values presented inferior hardness values (samples sets 8 and 9). Despite this, the variation in hardness values is rather low and similar to values measured in previous studies [22,55–57].

From comparing the Young's modulus obtained from the tensile test with that obtained from the ultrasound test, overall good agreement was found: less than 3% difference. Only for samples 3 and 6 was there was a higher error (~10%) due to the porosity, affecting dispersion and making the ultrasonic measurement difficult.

In order to better understand the differences in mechanical properties, fractography analysis was performed. The fracture surface of tensile specimens was analyzed in SEM, and typical samples are shown in Figure 6. The fractography results correlate with the elongation obtained (shown in Table 4 and depicted in Figure 5). Figure 6a, which presents the fracture images of sample 1b, shows that the fracture surface contains cavities and cracks in addition to unmelted powder particles. Dimples and facets can also be seen in the fracture surface, indicating that the fracture has mixed brittle-ductile behavior, matching the low elongation obtained (3.75%).

Figure 6b shows the fracture surface of sample 2a. It can be seen that the fracture features a classic cup-and-cone tensile fracture surface. Dimples are present at the fracture surface, indicating ductile behavior, correlating to the high elongation results obtained (15.9%). In addition, this sample also contains high micron-scale porosity (~50 μm) concentration, with cracks and cleavages also observed. While the pore population is substantial, their small volumes do not cause a significant decrease in density although the relative density measured (99.63%) was lower than ideal. The pore population did not cause low elongation values in these samples.

The fracture surface of sample 3a is presented in Figure 6c, showing that the fracture surface has large voids and a significant population of unmelted particles nested in the voids. This result is consistent with the low relative density values measured, 92.6%, and the porous surface. Facets are observed in the inset figure, which depicts the surface texture, indicating a brittle behavior, thus correlating with the low elongation obtained, 0.85%.

Figure 6d shows the fracture images of sample 4b. Overall, the fracture exhibits a cup-and-cone shape. However, large cavities are present at the fracture surface, and cracks were also observed in addition to dimples. These findings are consistent with elongation values indicative of a ductile fracture. Similar to sample 4b, the fracture surface of sample 5b also features the classic cup-and-cone tensile fracture surface, as can be seen from Figure 6e.

Moreover, some cracks can also be observed. Although the relative density is only 99.4%, the porosity is not noticeable. The fracture surface structure is mainly composed of dimples combined with the cup-and-cone shape observed, and the sample exhibit ductile behavior correlates to the elongation results obtained, 14.35%. The fracture surface images of sample 9a are presented in Figure 6f. From Figure 6, it can be seen that the fracture features the classic cup-and-cone tensile fracture structure. Many small-scale pores are noticeable at the fracture surface and cracks. Dimples structure can also be seen in the figure, indicating a ductile behavior, which correlates to the elongation results obtained, 16.41%. Despite the porosity and low relative density, a substantial elongation was obtained.

The findings from the fractography strengthen the assertion that density is a first-order factor of mechanical properties. While all samples had some degree of porosity, a highly common phenomenon in AM-PBF, the size and distribution of the pore population proved to be detrimental to mechanical properties.

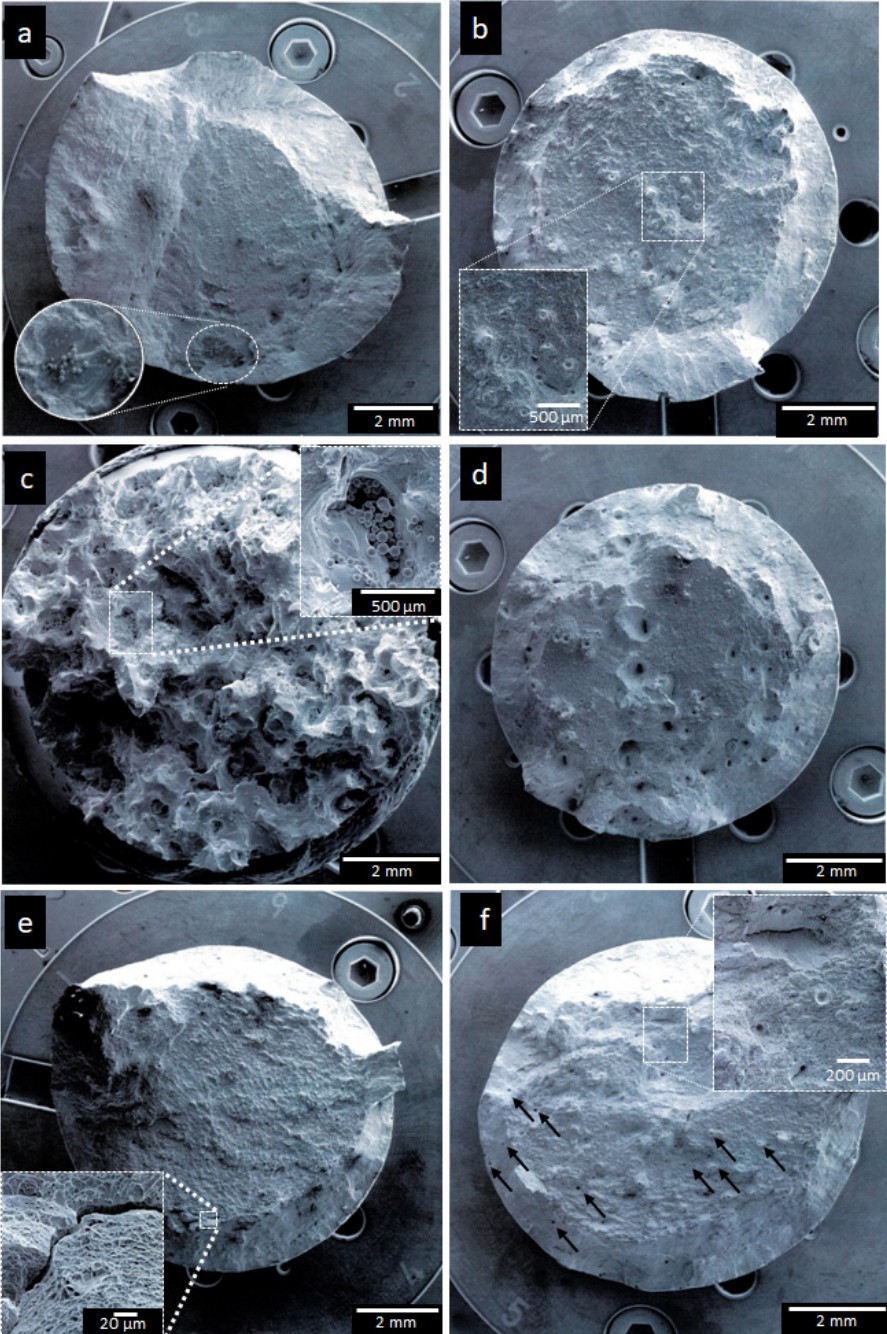

**Figure 6.** SEM tensile fracture surfaces of samples 1 (**a**), 2 (**b**), 3 (**c**), 4 (**d**), 5 (**e**), and 9 (**f**). Fractography reveals that all samples have a certain amount of pores. Their quantity and distribution vary within the samples, which affects the fracture mechanism.

Figure 7 illustrates the as-built microstructural features of representative samples 2, 5, 7, 8, and 9 obtained by scanning electron microscopy (SEM) imaging. Backscattered electrons (BSE) images were acquired within the x-y build planes (parallel to building direction, z). The bright and dark BSE contrast indicates the presence of a small volume fraction of the V-rich β phase and Al-rich α phase, respectively. The difference in the morphological characteristics of α and β phases between the samples is highlighted by the BSE images. The β length and width and α laths thickness are varied across the samples. When comparing the microstructure in Figure 7, it can easily be seen that samples 7 and 8 (Figure 7c,d) have fine features compared to samples 2 and 9 (Figure 7a,e), and an evident coarsening is observed in sample 5 (Figure 7b). In sample 5, continuous β ridges are

observed between adjacent α lathes, while in samples 2 and 9, the β phase breaks and is found as separate rods and "dot"-like structures with smaller spacings between them. This appearance of the "dot-like" structures' formation of the β phases and the decreased spacing is even more pronounced in samples 7 and 8. These differences in microstructure characteristics indicate different thermal regimes and cooling rates [7]. While in sample 5, the cooling rate was slow enough to allow the nucleation and growth for the formation of a lamellar structure and the observed coarsening, in samples 2 and 9, and even more so in samples 7 and 8, the cooling rates were too fast to allow growth of the phases, and therefore, the fine structure was obtained.

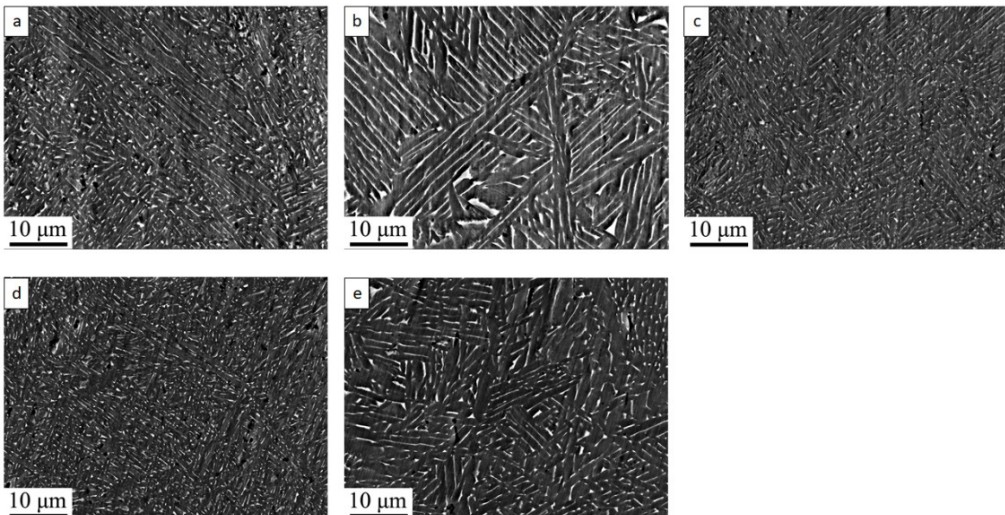

**Figure 7.** Backscattered electrons (BSE) scanning electron microscopy (SEM) images taken parallel to building direction of samples 2 (**a**), 5 (**b**), 7 (**c**), 8 (**d**), and 9 (**e**).

These microstructure results correspond to the mechanical properties results. As shown in Table 4, the highest strength was obtained in sample 8, followed by samples 2 and 9. Sample 5 exhibited the lowest strength of the samples in the SEM analysis. This samples' strength ranking is consistent with the microstructure feature-scale hierarchy observed.

*Taguchi Analysis*

In order to correlate the process parameters to the various properties measured, the main effect analysis was performed, including signal-to-noise (S/N) ratio calculations. The S/N ratio takes both the mean and the variability into account, as it measures the sensitivity of the quality characteristic to the noise factor. Hence, it can be defined as an inverse of the variance, and thus, the objective is to prefer the highest values regardless of the quality characteristics type (smaller-the-better, nominal-is-best, or larger-the-better) [58,59]. The main effect plots are shown in Figure 8. The horizontal dashed lines in the graphs represent the average S/N ratios. From Figure 8, it can be seen that, overall, the graphs have similar trends, which supports that process parameters have a similar effect on the material properties. The increase in beam current led to an increase in all properties, while the opposite trend was observed in the case of speed function. In terms of line offset, the highest properties were obtained at the second level, 0.22 mm. With regard to the focus offset, there is a trend difference within the properties, while the highest density and Young's modulus were obtained using 15 mA but with a slight difference; the highest YS and UTS were obtained using 45 mA but with a considerable difference.

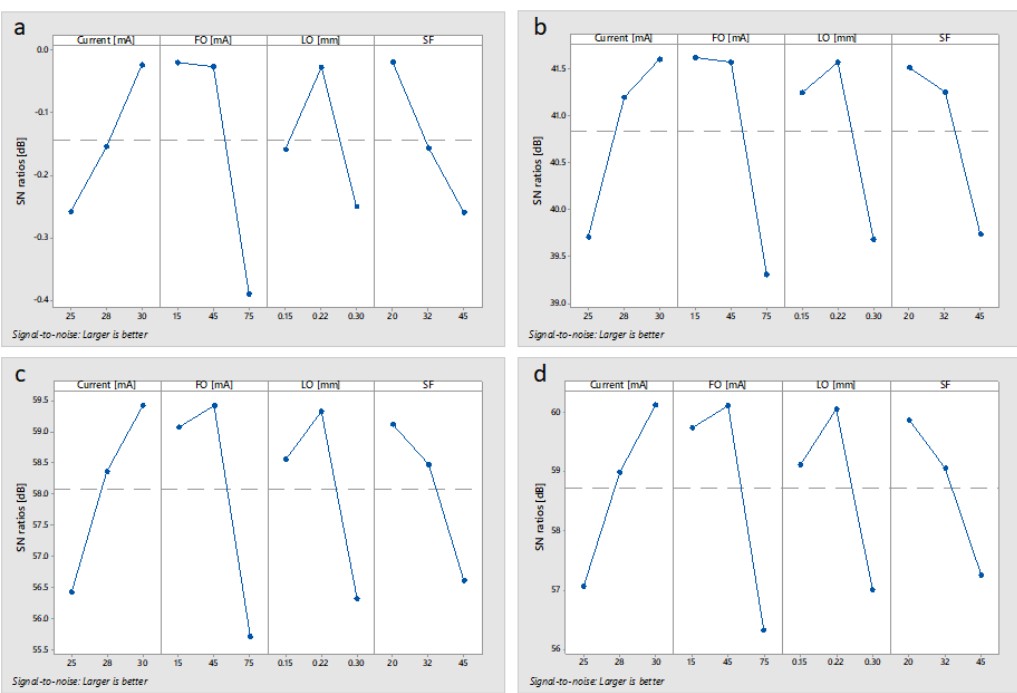

**Figure 8.** Effect of process parameters on density (**a**), Young's modulus (**b**), yield strength (**c**), and UTS (**d**).

The spread between the maximal and minimal values in each property indicates the impact extent of the parameter on the same property. In this case, similar trends of the impact extent of the parameters on the properties were obtained, while the focus offset fluctuations being the largest in all the various properties; i.e., it is the parameter that has the greatest effect on the material properties among the values of the parameters examined. This means that a slight change in focus offset value can cause a significant difference in properties.

Figure 9 shows the effect of line and area energy density on properties. As can be seen, using similar line and area energy densities yielded by different process parameter combinations, various properties were obtained, and conversely, similar properties were obtained using different energy densities. For example, samples 2 and 9, which have similar mechanical properties and even similar microstructure characteristics, were manufactured using different line and area energy densities, and samples 2, 6, and 7 were manufactured using similar line energy densities but presented different behavior. This result highlights that the parameters that dictate the quality of the final product are not a specific energy density but rather the combination of the parameters and their interrelationships. Hence, since almost identical properties can be obtained using different energy densities, it can be understood that energy density is a controversial examination criterion, and a specific value should not be taken as an absolute value [60–62]. There may be cases of different manufacturing processes in which the energy densities are similar, but different properties are obtained and vice versa. It is worth mentioning that the beam's diameter is not taken into account in the energy density calculations despite its significant effect on the melting and solidification process [63]. That could explain the difference in energy density values and be the factor that compensates for it. In addition, the energy density is expressed in energy per unit length or area, but there is no consideration for the time at which the same amount of energy is deposited. The duration of the irradiation has a decisive effect on the powder heating and the evacuation of the heat, and as a result, it affects the melt pool temperature and its dimension. Thus, for example, manufacturing using I = 10 mA and v = 1200 mm/s has a different effect than using I = 20 mA and v = 2400 mm/s although both have the same line energy density of 0.5 J/mm.

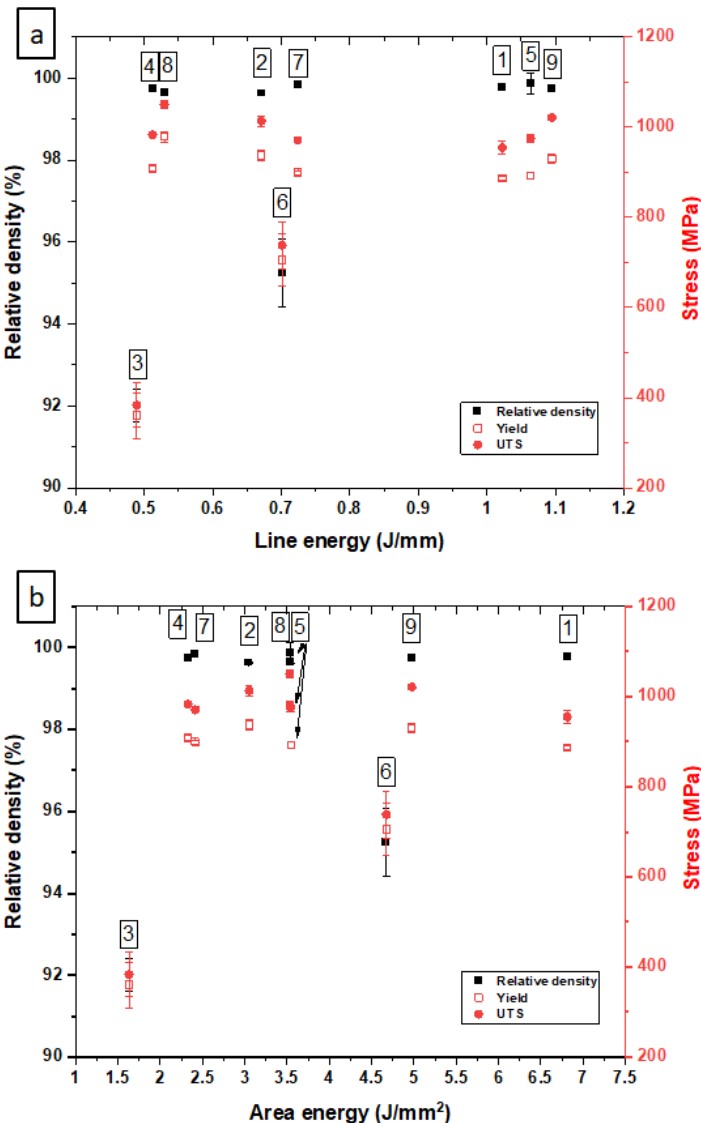

**Figure 9.** Effect of the line (**a**) and area (**b**) energy density on density, yield, and ultimate tensile strength.

As mentioned, the manufacturing parameters, expressed through energy density and its interrelationships, dictate the input energy levels to the powder bed, which control the melting of the raw material and determine the degree of melting (partial melting/full melting/over melting). This consequently affects the defect population in the product, such as porosity, delamination, balling, cracking, and so forth. In addition, the process parameters also control the melting pool features, the temperature gradient, and its cooling and solidification rates, which dictate the microstructure characteristics (nucleus size and shape, phases ratios, texture, etc.) [7]. Therefore, as long as the features of the melting pool and the solidification regime are sufficiently similar in different manufacturing processes, similar properties will be obtained regardless of the manufacturing parameters (as can be seen in samples 2 and 9).

Favored working parameters are those through which the best-desired properties are obtained, i.e., from Figure 8, and these are the parameters at the graph's peak. As noted before, the trends of the effect of process parameters on the various properties are similar except for the focus offset parameter, where the peaks for density and Young's modulus are achieved using a different value than the one that causes the highest material strength. To examine the effect of the manufacturing parameters on the above properties, an overall

evaluation criterion (OEC) was set to weigh all features into one score. The values are normalized, as each property has different units, and the OEC value is in percent. The normalization values are based on the highest values obtained in SLM, which constitute the highest of these properties [24] besides the relative density, which is already normalized. The OEC format is described below (Equation (5)), and the evaluation criteria, relative weights, and normalization values are described in Table 6. Since the examination of the parts is mainly on their functionality, the UTS and YS were taken into account with relatively high impact. Relative density affecting the parts' integrity was also given a high grade.

$$\text{OEC} = \frac{\rho}{\rho_b} \times \omega_1 + \frac{E}{E_b} \times \omega_2 + \frac{\sigma_y}{\sigma_{yb}} \times \omega_3 + \frac{\sigma_{max}}{\sigma_{maxb}} \times \omega_4 \tag{5}$$

where $\rho$, $E$, $\sigma_y$, and $\sigma_{max}$ are density, Young's modulus, yield strength, and UTS values that are measured, respectively; $\rho_b$, $E_b$, $\sigma_{yb}$, and $\sigma_{max,b}$ are the normalization values, and $\omega_1$–$\omega_4$ are the weight of each parameter.

**Table 6.** Evaluation criteria, relative weights, and normalization values. All characteristics are of the higher-the-better type.

| Criteria | Characteristic | Units | Rel. Weighting | Normalization Value | Quality Characteristic |
|---|---|---|---|---|---|
| Relative density | $\rho$ | % | 20% | 1 | Higher-the-better |
| Young modulus | $E$ | GPa | 15% | 125 | Higher-the-better |
| YS | $\sigma_y$ | MPa | 20% | 1330 | Higher-the-better |
| UTS | $\sigma_{max}$ | MPa | 45% | 1450 | Higher-the-better |

The overall evaluation grades of the samples are presented in Table 7 as well as the S/N ratio of each parameter set. Similar to the non-weighted results, sample sets 3 and 6 present the lowest overall scores, whereas sample set 3 exhibits significantly lower performance. On the other hand, the samples with the highest grades are of sample set 8. As for the S/N ratios, the larger the S/N ratio, the higher their average and also their repeatability, i.e., lower variance. However, the highest S/N ratio possible in this study is 40, which is the S/N ratio obtained for a score of 100 due to the logarithmic scale. It can also be seen from the S/N ratios that sample set 8 obtained the highest ratio, 38.3, with sample sets 2 and 9 being close with ratios of 38.1. The rest of the sample sets, 1, 4, 5, and 7, have similar grades slightly lower than those of parameter sets 2 and 9.

**Table 7.** Overall evaluation grade of the samples and their S/N ratio.

| Sample Set # | A (%) | B (%) | C (%) | D (%) | E (%) | S/N (dB) |
|---|---|---|---|---|---|---|
| 1 | 77.0 | 76.8 | 77.4 | 77.7 | 77.9 | 37.8 |
| 2 | 80.1 | 79.9 | 80.6 | 79.8 | 79.2 | 38.1 |
| 3 | 43.8 | 44.0 | 39.8 | 43.7 | 45.1 | 32.7 |
| 4 | 78.8 | 79.0 | 78.3 | 78.5 | 78.4 | 37.9 |
| 5 | 78.2 | 77.4 | 78.1 | 78.2 | 77.5 | 37.8 |
| 6 | 65.2 | 68.1 | 61.0 | 63.8 | 68.8 | 36.3 |
| 7 | 78.4 | 78.3 | 77.8 | 78.2 | 78.1 | 37.9 |
| 8 | 82.3 | 81.5 | 82.0 | 82.0 | 81.2 | 38.3 |
| 9 | 80.2 | 79.6 | 80.0 | 80.4 | 79.4 | 38.1 |

Similar to the non-weighted results, the main effect analysis was performed on the OEC results and is shown in Figure 10. The horizontal dashed line in the figure represents

the overall average of all S/N ratios. From Figure 10, it can be easily seen that the focus offset factor is spread over the broadest range of values; i.e., it has the greatest impact on the performance of the samples produced, similar to the non-weighted results. Since the objective is the highest S/N ratio possible, regardless of the quality characteristic, the optimal level for each parameter is the one that shows the highest S/N ratio of the three different levels. Thus, the optimal process parameters are beam current of 30 mA, focus offset of 45 mA, line offset of 0.22 mm, and speed function index of 20. Analysis of variance (ANOVA) was performed to evaluate whether the factors' effect on the process response is statistically significant and to measure its contribution to the performance. A confidence level of 99% was required in the ANOVA. The results of the ANOVA are shown in Table 8. It can be seen that all *p*-values are smaller than 0.01; thus, all factors can be considered statistically significant. The factor that was found to be most influential on the performance is the focus offset, while the speed function index has the smallest effect, as was also obtained from the main effect results.

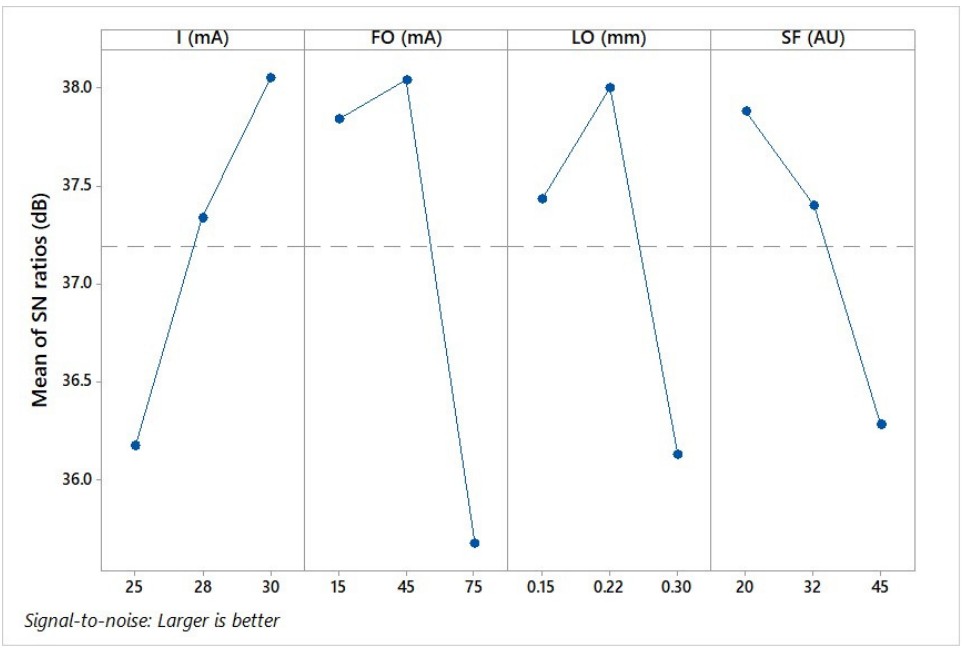

**Figure 10.** Main effect of each factor.

**Table 8.** ANOVA results.

| Factor | Degrees of Freedom | Sum of Squares | *p*-Value | Contribution (%) |
|---|---|---|---|---|
| Current | 2 | 1289.38 | <0.001 | 21.04 |
| Focus offset | 2 | 2618.17 | <0.001 | 42.73 |
| Line offset | 2 | 1311.02 | <0.001 | 21.40 |
| Speed function index | 2 | 846.48 | <0.001 | 13.82 |
| Error | 36 | 62.15 | | 1.01 |
| Total | 44 | 6127.20 | | 100 |

The performance under the optimal parameters was predicted using linear regression, and it was found that under these conditions, the OEC is expected to be 96.91. In order to confirm that the optimal parameters found do indeed improve the production process, the results are compared to the properties obtained by using Arcam's recommended parameters set: I = 28 mA, FO = 45 mA, LO = 0.22 mm, and SF = 32, shown in Table 9. According to the OEC, the overall grade of the properties obtained using Arcam's recommended parameters

is 79.1%. This overall grade is significantly lower than the predicted grade, 96.01, to be obtained using the optimal parameters found.

**Table 9.** Properties obtained using Arcam's recommended parameters.

| $E$ (GPa) | $\sigma_y$ (MPa) | $\sigma_{max}$ (MPa) | Density (g/cm$^3$) | OEC Grade (%) |
|---|---|---|---|---|
| 119.2 | 923.7 | 998.2 | 4.41 | 79.1 |

A comparison of the results obtained in the study shows that sample groups 2 and 9 have a similar overall performance to that obtained in samples manufactured according to parameters recommended by Arcam, while the overall performance of sample group 8 is even higher. The average OEC of samples groups 2, 8, and 9 are 79.92, 81.76 and 79.89%, respectively, while Arcam's is 79.09%. Therefore, even before using the optimal parameter set found, several manufacturing parameter combinations were found through which the product's overall performance can be improved. This is further evidence that through different combinations of manufacturing parameters, similar properties can be obtained.

The scores obtained in the study are based on a set of high-importance criteria by us, and Arcam can have a different set of criteria or different weights of importance. For example, productivity, expressed as time per layer, is a criterion that we do not address, and it is likely is of great significance for Arcam.

Using the optimal parameters set found, the predicted overall performance is 96.91%, a significant improvement compared to the result obtained using the parameters set recommended by Arcam and also to the results obtained in the various samples' groups. As can be seen from the experimental design table shown in Table 8, none of the experiments were conducted using the optimal parameters. The closest combination of parameters to the optimal one found is that in experiment 9, in which all parameters are the same except for the focus offset value. This result is interesting since while the parameters of experiment 9 are closest to those found to be optimal, the parameters of experiment 8 obtained higher grades than those obtained by experiment 9. This can be explained by the fact that, as mentioned above, the only difference between the parameters of experiment 9 and the optimal parameters found is in the level of the focus offset parameter, 75 mA versus 45 mA, respectively. From Figure 10, it can be seen that when the focus offset was 75 mA, the average grade obtained was the lowest and by a considerable difference from those obtained when it was 15 or 45 mA. Figure 11 shows the focus offset's effect on the process response, and it can be easily seen that the focus offset value of 75 mA causes considerable variation, while in the other two levels, the results are relatively repetitive. In addition, it can be seen that out of the three highest performances, namely experiment parameters 2, 8, and 9, two of them, the highest and third place, were performed using focus offset 45 mA. Thus, changing the focus offset value in the parameters set 9 from 75 to 45 mA will probably improve the process response.

Moreover, from a comparison of the optimal parameters set found to the parameters recommended by Arcam, it can be seen that a match was obtained in two values of the optimal conditions to those recommended by Arcam, namely in the focus offset and the line offset, 45 mA and 0.22 mm, respectively. As for the other two factors, the optimal beam current is a bit higher than Arcam's recommendation, 30 versus 28 mA, while the optimal speed function is a bit smaller, 20 versus 32 (i.e., slower beam speed). Therefore, slightly increasing the beam current while slowing it down is expected to improve the final product performance.

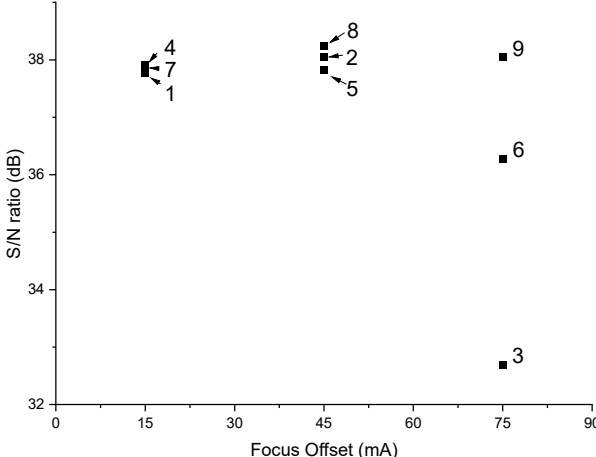

**Figure 11.** Focus offset effect.

## 4. Conclusions and Summary

In this work, the effect of process parameters on the properties of as-built Ti-6Al-4V fabricated by EB-PBF was investigated. The study examined the four most dominant manufacturing parameters in the melting phase: beam's current, velocity, focus offset, and line offset, at three levels each. The Taguchi method was used to reduce the number of experiments, which can be used to study the effect of the parameter levels using only 9 experiments rather than the possible 81 combinations. The samples underwent tensile tests, hardness measurements, density, surface roughness, ultrasonic, and fractography analysis. According to an evaluation criterion composed of density values, yield and ultimate tensile strength, and the modulus of elasticity, each parameter combination was graded with a score describing the overall material performance. Using various parameter combinations, a wide range of properties was obtained from porous materials with 8% porosity to almost fully dense materials with density higher than 99.9%, a range of yield and ultimate tensile strength values, and brittle samples with less than 1% elongation to ductile samples with an elongation greater than 16%. A correlation between mechanical properties and microstructure features was observed. Most of the samples obtained in the study meet the ASTM standard for PBF and even exceed the threshold significantly except for sample group 1, which does not meet elongation criteria, and sample groups 3 and 6, whose characteristics do not meet the standard at all. Although surface roughness was not included in the evaluation criterion, most of the samples obtained in the study had relatively low roughness.

It was found that samples manufactured under certain parameter combinations yield properties similar to those obtained using the machine manufacturer's recommended parameter combination. Furthermore, it is possible to achieve even better parameters when considering specific characterizations mentioned above. By using evaluation equations, it is possible to pre-design the desired properties from the material and, as a result, obtain the production parameters required to obtain the desired properties. It was found that within the range of the values of the parameters tested, the most influential parameter is focus offset, while the speed function index is the parameter with the lowest impact. Due to the large number of parameters that affect the additive manufacturing processes, Taguchi's method has been found to be effective and beneficial in reducing the number of experiments and finding process conditions that improve the material's properties obtained using the standard parameters and have the potential to improve it even further. It was established that DOE is a useful tool for developing an optimal parameter set for new material.

**Author Contributions:** Conceptualization, N.F., S.H. and E.T.; methodology, N.F., S.H., D.B., Y.I.G. and E.T.; validation, D.B. and Y.I.G.; formal analysis, G.M.G. and S.S.; investigation, D.B., Y.I.G., S.S. and G.M.G.; resources; N.F., S.H. and E.T.; data curation, D.B., Y.I.G., S.S., G.M.G. and M.C.; writing—original draft preparation, D.B. and Y.I.G.; writing—review and editing, N.F., S.H. and E.T.; visualization, D.B., Y.I.G. and E.T.; supervision, N.F., S.H. and E.T.; project administration, N.F., S.H. and E.T.; funding acquisition, N.F., S.H. and E.T. All authors have read and agreed to the published version of the manuscript.

**Funding:** This research was funded by Pazy foundation, grant number 2020-ID147.

**Data Availability Statement:** Available upon request.

**Conflicts of Interest:** The authors declare no conflict of interest.

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
