# Peer review of "A Design of Experiment Approach for Development of Electron Beam Powder Bed Fusion Process Parameters and Improvement of Ti-6Al-4V As-Built Properties"

_jmmp, doi:10.3390/jmmp6040090_

Round 1
Reviewer 1 Report
Make a more specific conclusion in the abstract. The conclusions are a bit general and it is not clear what specifically had the most profound impact on the final outcome.
Keep the font consistent. It changes size throughout the text. Example, beginning of the second paragraph in intro.
Please refer to the following as they focus on powder bed process:
https://books.google.com/books/about/Additive_Manufacturing_of_Metals.html?id=Pe93zQEACAAJ
Please use the correct terminology. E.g. Hatching distance should be replaced by hatch spacing.
How did you select the parameters range? For example, beam current is varied between 25-30 mA. What is the possible range and why is this range selected?
Please provide the ultrasound equations used to characterize the samples. Specifically the equations from reference 43 and 44 which was used to calculate the mechanical characteristics such as Youngs modulus, shear modulus etc.
Did you compare the results of tensile tests with the ultrasound measurements? In other words, the young’s modulus obtained from tensile test with the Youngs modulus obtained from the ultrasound test??
Author Response
The authors wish to thank the valuable feedback. We have carefully considered the comments and suggestions, all the modifications made in the paper are highlighted in red using "Track Changes". Below, comments and suggestions are detailed, point-by-point.
Response to Reviewer 1
- Addition to the conclusion in the abstract – corrected. Referred to in lines 28-31
- Regarding font consistencies – corrected. See lines 12, 14, 19, 49
- The suggested source was added and referred to in line 60
- The hatching distance was replaced with the suggested terminology, hatch spacing – line 80
- Since our reference point was the existing set of parameters recommended by the manufacturer (Arcam), the parameter range was determined according to it, with Arcam's parameters being in the center of the process parameters range. These parameters ranges were determined in this way in order to ensure that measurable samples could be obtained in each combination. Since the deposited energy depends on each tested parameter, it is impossible to test vast working ranges for fear of extreme points with very high energy which will cause swelling and stop the process during it, or alternatively, a lack of energy that will prevent melting. Therefore, a working window was tested, which allows obtaining samples from each parameter combination but on the other hand, to obtain samples with different properties. An explanation was added in lines 134-136
- Revised, please see lines 193 & 199-203
- Yes, we did compare the Young's modulus obtained from the tensile test with the one obtained from the ultrasound test and in overall, we found good agreement, less than 3% error. Only for samples 3 & 6, the porous ones, there was a higher error (~10%) due to the porosity causing dispersion and affecting the ultrasound measurements. (See reference in the text in lines 280-283).
Reviewer 2 Report
This paper studies the effect of process parameters (beam current, velocity, focus offset, and line offset) on the defect, surface roughness, microstructure, and tensile properties of selective electron beam melting Ti-6Al-4V alloy. The quality of the manuscript should be further improved. The comments are as follows.
1) There are many literatures regarding the effect of process parameters on the properties of selective electron beam melting Ti-6Al-4V alloy (some Refs are listed below). Also, some paper have used the Taguchi approach to optimize the build quality of AM Ti6Al4V alloy. Therefore, I think the novelty of this manuscript is not enough.
Microstructures and mechanical properties of Ti6Al4V parts fabricated by selective laser melting and electron beam melting. Dec 2013 JOURNAL OF MATERIALS ENGINEERING AND PERFORMANCE 22 (12), pp.3872-3883
Microstructure, anisotropic mechanical properties and very high cycle fatigue behavior of Ti6Al4V produced by selective electron beam melting. Aug 2021 METALS AND MATERIALS INTERNATIONAL 27 (8), pp.2550-2561
Influence of Inherent Surface and Internal Defects on Mechanical Properties of Additively Manufactured Ti6Al4V Alloy: Comparison between Selective Laser Melting and Electron Beam Melting. Apr 2018 MATERIALS 11 (4)
Effect of energy input on microstructure and mechanical properties in EBSM Ti6Al4V. 2018 | MATERIALS AND MANUFACTURING PROCESSES 33 (15), pp.1708-1713
2) Details about the selective electron beam melting experiments should be provided. Electron beam scanning strategy? Vacuum degree? Pre-heating temperature? Relationship between tensile test direction and build direction?
3) The scale bar of Figure 3 is missed. And, when study the relative density of as-built sample, the OM images showing the defect characteristics should be provided to better research the densification behavior.
4) The definition of Figure 6 is too low.
5) The author said that “it can easily be seen that samples 7 and 8 (Figure 7c and d) have fine features compared to samples 2 and 9 (Figure 7a and e), and an evident coarsening is observed in sample 5 (Figure 7b)”. The related quantitative analysis should be provided.
6) The microstructure and tensile properties of as-built Ti6Al4V alloys are similar to those in previous literatures. What is the new discovery of this paper?
7) “Using the optimal parameters set found, the predicted result is 96.91%...”. This results is very bad as in many literatures the relative density of as-built Ti6Al4V is as high as 99.5% and more. So, what is the meaning of using the DOE method?
Author Response
Response to Reviewer 2
- Indeed, many studies dealt with the effect of the process parameters on microstructure, mechanical properties and defects. With regard to the specific proposed refs, two of them compared obtained properties of SLM vs. EBM, and a third source examined the effect of build orientation on static and dynamic mechanical properties. So, since their methodology of studying the AM products was implemented in this current study, they are now cited in the introduction section, (please see lines 62-64). Nevertheless, these works were performed without changing the manufacturing parameters, differing from our work. One source tested the effect of the energy input on the resulting properties, and he did so by changing the beam current. This single change in the printing theme led us to the comprehension that a multi-parameters study should be of interest. This specific paper was added to our reference list, implemented in the motivation part, please see line 96. Most of the studies in the literature have examined the effect of a single parameter on the obtained properties or through the creation of process maps. As known, a change in one of the fixed parameters in the experiment can cause changes in the trends obtained. In our study, we examine several parameters simultaneously, which allows for obtaining trends that depend on all the parameters and not on individual parameter. In addition, the purpose of this article is to propose a working method that allows the improvement of the existing properties. Furthermore, since today, most of the articles dealing with the development of manufacturing parameters of new materials do so by creating process maps, our work shows that the suggested method is an effective tool for the development of manufacturing parameters of new materials (see two last paragraphs in the introduction – lines 94-122)
- Thanks for the remarks, the details were added. (lines 153-154)
- The scale bar of figure 3 was added to the figure. This work focused on developing a DOE that improves manufacturing parameters and sets new parameters. Therefore, no in-depth study of the solidification mechanisms was performed and no analysis of cross-sectional images of the samples with defects was performed. This topic will be included in part of a follow-up study, which examines more aspects in-depth.
- An additional description has been added to Figure 6 – lines 330-332
- While comparing quantitative values rather than comparing qualitatively is always preferred, the Windmanstatten microstructure greatly hinders the traditional grain size approximation methods. Furthermore, even using methods such as lath width measurements are not easily applied to such specimens. As this work focuses more on the development and proof of the conceptual experiment design, we believe the qualitative comparison should suffice.
- In this work, the results obtained are similar to those reported in the literature and even those obtained using the parameters recommended by Arcam, and this is the significant discovery, that the proposed DOE does not cause distortions and obtain inferior results. That is, the proposed DOE can be used as an effective way to develop manufacturing parameters for new materials, for which the manufacturing process has not yet been developed. Using this DEO the development stages can be shortened compared to the currently common way of creating process maps using the full factorial method.
- Samples density achieved in this study is detailed in table 4, with 99.84% relative density. Regarding the result of 96.91%, we should rephrase and clarify that we meant that using the optimal parameters, the predicted grade of weighting the total properties is 96.91%, not the relative density. Hence this is a significant improvement relative to the current state. See clarification in line 496.
Round 2
Reviewer 1 Report
The authors have addressed all the comments adequately.
Reviewer 2 Report
I think the quality of the revised manuscript is good.